# The Effect of the Overexpression or Addition of IGF1 and FGF2 in Human Chondrocytes Included in a Fibrin Matrix and Cultivated in a Dynamic Environment

**DOI:** 10.3390/polym16141968

**Published:** 2024-07-10

**Authors:** Jorge Lara-Arias, Victor Manuel Peña-Martínez, Luis Alejandro Rodriguez-Corpus, Viktor J. Romero-Díaz, Eduardo Álvarez-Lozano, Herminia G. Martínez-Rodríguez

**Affiliations:** 1Orthopedics and Traumatology Service, Hospital Dr. José E. González, Universidad Autónoma de Nuevo León, Av. Dr. José Eleuterio González 235, Mitras Centro, Monterrey 64460, NL, Mexico; jorgelara77@gmail.com (J.L.-A.); doctorviko@hotmail.com (V.M.P.-M.); luisalecorpus@gmail.com (L.A.R.-C.); 2Histology Department, Facultad de Medicina, Universidad Autónoma de Nuevo León, Av. Dr. José Eleuterio González 235, Mitras Centro, Monterrey 64460, NL, Mexico; vikromero@email.com; 3Biochemistry and Molecular Medicine Department, Facultad de Medicina, Universidad Autónoma de Nuevo León, Av. Dr. José Eleuterio González 235, Mitras Centro, Monterrey 64460, NL, Mexico

**Keywords:** tissue engineering, dynamic culture, transfected chondrocytes, growth factors

## Abstract

Hyaline cartilage is a highly specialized tissue. When injured, its repair capacity is low, which results in the massive destruction of the articular surface. Using tissue engineering and genetic engineering techniques, it is possible to provide a suitable microenvironment providing chondrocyte growth factors involved in the development of hyaline cartilage proteins, as well as cell proliferation and differentiation. Our aim was to stimulate the synthesis of an extracellular matrix via the chondrocytes included in a fibrin matrix through the addition or overexpression of IGF1 and/or FGF2, while maintaining a constant agitation of the culture medium. Collagen type II and glycosaminoglycans increased during the entire incubation time. In contrast, collagen type I decreased its expression under the same culture conditions, transfecting or supplementing growth factors to chondrocytes. However, chondrocytes that were not transfected or supplemented showed a general increase in the proteins analyzed in this study. The presence of IGF1 and FGF2 increased the protein synthesis of the hyaline cartilage, regardless of which one was the source of growth factors. Continuous agitation using the spinner flask allows for the adequate nutrition of chondrocytes included in the fibrin matrix. However, they require growth factors to up-regulate or down-regulate collagenous proteins.

## 1. Introduction

In recent years, the use of different growth factors to stimulate the synthesis of extracellular matrix in different cell types has been more intensively explored. Chondrocytes remain the ideal cell units for articular cartilage regeneration. The growth factors that stimulate changes in chondrocytes are IGF-1, FGF2, and TGFß1 [1,2]. Some have even been produced in genetic engineering for cartilage tissue injuries [3]. TGFß1 is known to play a crucial role in chondrocyte proliferation and the synthesis of extracellular matrix components. However, we focused on IGF1 and FGF2 due to their specific roles in promoting collagen type II synthesis and glycosaminoglycan production, which are essential for hyaline cartilage formation. In the present study, we compare the effects on extracellular matrix production in the chondrocytes included in fibrin by adding purified recombinant IGF1 and/or FGF2 to the culture medium and using a plasmid system that enables the continuous expression of these growth factors for at least 30 days. Furthermore, we employ a system with continuous stirring, which stimulates the flow of nutrients through the fibrin matrix, allowing the availability of chondrocytes present within the matrix.

## 2. Materials and Methods

### 2.1. Chondrocyte Culture

Primary cultures corresponding to patients seeking autologous chondrocyte implantation were used. Cell culture was performed according to the protocols established in the Bone and Tissue Bank of the Orthopedics and Traumatology Department, University Hospital “Dr. José E. Gonzalez.” The osteochondral biopsy was macerated into small pieces and subjected to several cycles of enzymatic digestion using 2.5% trypsin/EDTA (GIBCO-BRL Life Technologies, Grand Island, NY, USA) under sterile conditions. Subsequently, digestion was performed using collagenase type II (GIBCO-BRL Life Technologies) under continuous agitation at 37 °C. The cells contained in the supernatant were pelleted via centrifugation at 1000 rpm for 10 min. We used six-well plates for the cell culture, and cells were plated at a density of 1 × 105 cells per well (NUNC™, Rochester, NY, USA). The cell culture medium was Opti-MEM I (GIBCO-BRL Life Technologies), supplemented with 10% fetal bovine serum (FBS) (GIBCO-BRL Life Technologies). The chondrocyte cultures were maintained in an atmosphere of 5% CO_2_ and 100% relative humidity.

### 2.2. Plasmid Vectors and Transfection

We digested the pSPORT1 LIFESEQ780074 commercial vector (Open Biosystems, Inc., Huntsville, AL, USA) carrying the cDNA of the human insulin-like growth factor 1 gene using the NotI/SalI restriction enzymes. A 780 bp fragment was released and subsequently subcloned into the pCMV-Sport6 vector, placing the expression of IGF-I under the control of the cytomegalovirus promoter. The pCIneo plasmid, carrying the cDNA of the fibroblast growth factor 2 gene (FGF-2), was provided by Dr. Claudia Heilmann, University of Freiburg, Germany. This clone was characterized by digestion with the NheI/XbaI restriction enzymes, which resulted in the release of an 868 bp fragment. Both clones were also characterized by sequencing. We used the vector pGREEN LANTERN™-1 (Invitrogen™ Life Technologies, Carlsbad, CA, USA), which expresses green fluorescent protein as a reporter gene. Lipofectamine™ 2000 (Invitrogen™ Life Technologies) was used for transfection. The conditions for co-transfections were established according to the manufacturer’s instructions (5 μL Lipofectamine™ 2000/μg DNA), and we used a cell confluence of 90%.

### 2.3. Adding Growth Factors to the Culture Medium

Supplementations with 5 μg/mL of IGF-1 (Sigma, St. Louis, MO, USA) and FGF-2 (Sigma, St. Louis, MO, USA) were conducted separately and in combination (10 μg/mL total) every third day when the fresh culture medium was changed.

### 2.4. Chondrocyte Inclusion and Dynamic Culture

At 24 h post-transfection, the chondrocytes were detached from the culture dish via digestion with 2.5% trypsin/EDTA. Both transfected and non-transfected chondrocytes were embedded in three-dimensional media using a fibrin-based commercial adhesive (Baxter, Vienna, Austria). Following the manufacturer’s instructions, chondrocytes were suspended in fibrin before mixing them with the fibrinogen and thrombin provided with the kit. After solidification, the implant was cut into small 3 mm [3] cubes and grown in a spinner flask with 25 mL of complete medium (Optimem I/SFB 10%/gentamicin 5 μg) under constant stirring and controlled atmospheric conditions (5% CO_2_, 37 °C and humidity 100%). The medium was changed every three days. Triplicate samples were removed every seven days for four weeks. Molecular, histological, and immunohistochemical tests were performed on the medium. 

### 2.5. Western Blotting

Proteins were precipitated from the culture medium through methanol–chloroform extraction and separated on a 12% polyacrylamide gel under denaturing conditions. For protein detection, we used the AmpliCruz ™ Western Blot Signal Enhancement System (Santa Cruz Biotechnology, Inc., Dallas, TX, USA) commercial kit, following the manufacturer’s instructions. We used the following primary antibodies: FGF basic antibody (ab10420, 1:1000) and IGF1 antibody (ab9572, 1:3000) from Abcam plc (Cambridge, MA, USA). Rabbit IgG secondary antibody H&L (ab6721) from Abcam plc (Cambridge, MA, USA) was used as a secondary antibody. To visualize the proteins, we used the Luminol Chemiluminescence Reagent kit (sc-2048, Santa Cruz Biotechnology, Inc.). β-Actin (Invitrogen™ Life Technologies. Carlsbad, CA, USA) was used as a loading control to normalize the protein amounts loaded in each lane, ensuring that observed signal variations were not due to differences in protein loading.

### 2.6. RNA Extraction and RT-PCR

RNA was obtained from explants collected every seven days following the extraction protocol accompanying the Purescript ^®^ kit manufactured by Gentra Systems, Inc. (Minneapolis, MN, USA). Specific primers for the detection of collagen type I, type II, and GAPDH were designed using Primer3 software (2.6.0 version). The kit used for RT-PCR was the Advantage^®^ RT-for-PCR Kit (Clontech, Takara Bio Inc., Shiga, Japan). Reactions were performed in the GeneAmp PCR System 9700 from Applied Biosystems (Foster, CA, USA). The conditions for the amplification of collagens I and II were the following: 94 °C for 5 min; 35 cycles of 94 °C for 30 sec, 68 °C for 30 s, and 72 °C for 1 min; and 72 °C for 3 min. The PCR products were visualized on a 1.5% agarose gel (Invitrogen™ Life Technologies), and the obtained bands were observed with a UV transilluminator (Gel Documentation System 1000, Bio-Rad, Mexico City, México). The sequences for the primers were as follows: Collagen Type I: Forward 5′ATGACGTGATCTGTGACGAGAC 3′, Reverse 3′GCAGCACCAGTAGCACCATCAT 5′ (Product Length: 737 bp); Collagen Type II: Forward 5′AGACATCAAGGATATTGTAGG 3′, Reverse 3′CAGCTTCACCATCATCACCAG 5′ (Product Length: 403 bp); GAPDH: Forward 5′AAGATGGCCCAGGAGAACCCCAAG 3′, Reverse 5′TAATCCTTCATGTGCACCGCCCTG 3′ (Product Length: 890 bp). Normalization was performed using the constitutively expressed gene GAPDH, and the analysis was carried out via densitometry.

### 2.7. Histology and Immunohistochemistry

The explants were fixed with 4% paraformaldehyde and dehydrated with ethanol at subsequent concentrations of 85%, 98%, and 100%. Next, the explants were placed in xylol for 10 min and embedded in paraffin melted at 60 °C. Finally, each piece was placed in a mold and cut with a microtome (3 μm). These samples were fixed in albumin and xylol for 10 min, then rehydrated in decreasing concentrations of ethanol, i.e., 100%, 98%, and 85% (20 times each). This process ended with the placement of the slices in water for full rehydration. 

We used the Safranin O technique to detect glycosaminoglycans (GAGs) produced by chondrocytes in fibrin. In addition, specific primary antibodies against collagen I and collagen II (1:50) (Abcam plc, Cambridge, MA, USA) were used for the detection of collagen fibers arranged in the new extracellular matrix of the fibrin/chondrocyte scaffold. We used the mouse ABC Staining System (sc-2017) and the rabbit ABC Staining System (sc-2018) from Santa Cruz Biotechnology, Inc. The slides were treated with decreasing concentrations of ethyl alcohol (from 100 to 70%), treated with xylol for 10 min, and then fixed with Entellan^®^ (Merck, Darmstadt, Germany). Coverslips were placed on the slides, which were observed under an inverted microscope attached to a computer system for image capture. 

### 2.8. Image Analysis

All images from RT-PCR and immunohistochemistry (IHQ) experiments were analyzed with Image-Pro Express software 6.2 (Media Cybernetics, Inc., Rockville, MD, USA), which estimates the standard optical density by determining the amount of material in an object by measuring the amount of light passing through the object. The results were expressed as the means ± standard deviations (SD) (n = 3) and evaluated using a 95% confidence interval for statistical significance (*p* < 0.05). The quantification of staining intensities was performed via densitometry using the optical density formula to normalize the intensity of the staining: Optical Density (x, y) = −log (Intensity (x, y)–Black/Incident−Black), where Intensity (x, y) is the pixel intensity at position (x, y), Black is the intensity generated when no light passes through the object, and Incident is the intensity of the light incident on the sample. This formula assumes an exponential decay of light within the object, allowing for the accurate quantification of staining density. 

### 2.9. Statistical Analysis

Data are presented as means ± standard deviations (SD). Statistical differences between groups were analyzed using one-way ANOVA followed by Tukey’s post hoc test. A value of *p* < 0.05 was considered statistically significant. All experiments were repeated at least three times to ensure the reliability of the results.

## 3. Results

We were able to detect the expression of IGF1 and FGF2 in chondrocytes transfected and included in the fibrin (Figure 1) throughout all 30 days of the assay and under all the different conditions used. When analyzing the expression of extracellular matrix proteins characteristic of fibrous or hyaline cartilage, such as collagen types I and II, respectively (Figure 2A,B), we noted a significant difference in the production of type I collagen fibers in cells co-transfected with both growth factors (25.09 ± 1.53), as compared to the expression of either alone (FGF2: 31.33 ± 0.6; IGF: 131.92 ± 1.63). This difference was already detectable in the first week of expression. As the assay continued for the entire 30 days, we observed that the expression levels of collagen I gradually decreased. By the fourth week, there was no significant difference between the levels of collagen I across the different conditions. 

We found no significant difference between adding and transfecting IGF1 (Figure 3). In general, collagen type I gradually decreased from the first to the fourth week, contrary to what was observed with collagen type II, where the production of this protein was higher at the endpoint under both conditions (Figure 3A,B). GAGs showed similar behavior to collagen type 2. An increase was observed in the synthesis of these compounds. Supplementing IGF1 (Figure 3), the values for type I collagen were 36.14 ± 2.74 in the first week and 27.61 ± 2.19 in the fourth week; these values represent the optical density (OD) measurements from the immunohistochemical staining. Similar results were observed in transfected chondrocytes included in the fibrin: in the first week, the densitometric values were 35.39 ± 1.01 and in the fourth week, 29.01 ± 2.08, suggesting that IGF-1 may influence the down-regulation of collagen type I. The length of the fibers of collagen type II throughout the three-dimensional matrix increased as the days passed. Under different stimulus conditions, the significant difference between the first and fourth week of production of collagen type II was confirmed. 

During the four weeks of the study, the supplementation or overexpression of FGF2 (Figure 4) did not cause a difference between the two events. The behavior observed for collagen types I and II was very similar to the results reported for IGF1. GAGs were also positively affected under the conditions that were studied for the chondrocytes included in the fibrin matrix (Figure 4A,B). The control samples consisted of chondrocytes embedded in a three-dimensional fibrin matrix without any supplementation or addition of growth factors.

Transfecting both growth factors individually, a significant difference was observed for the expression of collagen type I between the first and fourth weeks, so that the percentage of the area occupied for samples where FGF2 was overexpressed was 36.11 ± 1.58 and 28.07 ± 1.2.2, respectively, suggesting that FGF2 stimulation may influence the synthesis of this protein. When the same recombinant protein was supplemented to the medium, a similar effect was observed: the collagen type I production decreased as the days passed for this trial. It was observed that the collagen type II fibers increased gradually and continuously in both stimulus conditions. The densitometric values to estimate the percentage of area that is positive for collagen type II showed that FGF2 supplementation for the first week revealed that 30.17 ± 2.62 percent of the implants were occupied by this protein and 30.26 ± 1.61 percent of the implants were occupied by collagen II when chondrocytes were transfected (Figure 4). For the fourth week, the implant was occupied by collagen type II in a 60.94 ± 4.81 percent when the stimulus was provided by the overexpression plasmid of FGF2, and 42.52 ± 4.37 when the stimulus was due to the addition of this recombinant protein to the medium. Collagen type II (Figure 5) for the co-transfection of both growth factors was synthesized in a higher proportion in the first week (31.27 ± 1.41) compared to the stimulus provided on this protein by FGF2 (19.51 ± 0.6) and 20.31 ± 1.03 that IGF1 generated. There was no significant difference in the levels of collagen type II when growth factors were co-expressed simultaneously during the 30 days; in addition, for the growth factors expressed individually, the synthesis gradually increased, observing a significant difference in the levels from the first week to the fourth week. However, there was no statistical difference between them at the set time. 

When both growth factors were transfected, it was found that the percentage of positive areas of type II collagen was statistically significant during the first week (40.53 ± 1.77) with respect to the areas occupied by the collagen type II when growth factors were transfected individually (IGF1 32.09 ± 2.02/FGF2 30.26 ± 1.61) (Figure 3 and Figure 4). However, we found no significant difference with respect to the positive area during the fourth week, regardless of the chondrocytes embedded in the fibrin matrix being transfected or not. Collagen type I was found with decreasing values of 35.31 ± 3.99 in the first week and 27.24 ± 1.62 in the fourth week, when the stimulus was provided by both growth factors supplemented to the culture medium. For all assays, GAGs progressively occupied areas through collagen fibers. Positive regions were observed for collagen type II in greater proportion during the addition to the medium (61.74 ± 5.93) or during the transfection (59.14 ± 4.81) of FGF-2/IGF-1, compared with samples that were only stimulated by culture medium without supplement of any recombinant protein.

Concerning the dynamic stimulus (Figure 6), the observed results were similar with respect to the formation of extracellular matrix based on type II collagen and GAGs; in fact, the areas occupied by these proteins were 55.27 ± 1.97 and 57.68 ± 2.51, respectively, for the fourth week. However, the fibers of collagen type I increased in their distribution through the fibrin. During the first week, the positive areas represented 31.38 ± 3.99 and in the fourth week; when the assay ended, the area occupied by this protein was 55.14 ± 2.57 percent. Figure 7 and Figure 8 provide a good example of the affinity of specific antibodies for the detection of collagen types I and II, which indicates the quality of the new matrix formed under the stimulation of growth factors IGF1 and FGF2, alone or together, and when they were added or overexpressed by a plasmid vector. When FGF2 and/or IGF1 are present, a reduction in the synthesis of collagen type I can be appreciated. In the fibrin matrix are fibers of collagen types I and II and glycosaminoglycans. At 1 week, the Safranin O staining indicates the initial deposition of GAGs in the fibrin matrix. Over the 4-week period, there is a noticeable increase in the intensity and distribution of GAG staining, particularly in the samples with combined hFGF2 and hIGF1 treatment (panels I–L). The differences in GAG distribution and staining intensity between 1 week and 4 weeks highlight the enhanced matrix production over time due to the presence of the growth factors. (Figure 9). In general, the reductions in collagen type I and the gradual increases in collagen type II and glycosaminoglycans were similar under different culture conditions.

## 4. Discussion

Damages to articular cartilage represent a clinical pathology commonly encountered during medical practice. The exposure of cartilage to any agent that can cause any injury may suppress proteoglycan synthesis and stimulate the process of articular cartilage degeneration [4]; these defects can be filled with fibrocartilage that provides tissue with a different strength [5]. There are different growth factors that stimulate growth and even promote hyaline cartilage repair [6]. In this study, we focused on IGF1 and FGF2 to understand their combined effects on the synthesis of collagen type II and glycosaminoglycans in human chondrocytes. While TGF-β1 is a significant cytokine known to enhance chondrocyte proliferation and extracellular matrix production, it was not included in our experimental design. Previous studies have demonstrated that TGF-β1 can lead to different outcomes in cartilage formation under various conditions. Specifically, TGF-β1 signaling through Smad2/3 has been shown to inhibit chondrocyte terminal differentiation, potentially leading to the formation of fibrocartilage rather than hyaline cartilage [7]. Therefore, our research specifically targeted IGF1 and FGF2 to explore their potential in promoting hyaline cartilage characteristics, which are critical for articular cartilage repair. FGF2 is a mitogenic factor for articular chondrocytes [8,9] and stimulates or inhibits the deposit of proteoglycans depending on its concentration and the culture system employed [8,9,10,11]. Although FGF2 has an average life of less than 24 h [12], G. Kaul et al. tested the hypothesis that the overexpression of FGF2 transfections using chondrocytes embedded in alginate promotes the repair of full-thickness defects of articular cartilage in rabbits and releases protein at least one month in vitro [13]. In the present study, we could detect the presence of FGF2 in the culture medium for up to 30 days so that we could perform in vivo assays using transfected chondrocytes explants included in fibrin. Marja et al. examined the effect of FGF2 on levels of procollagen αI mRNA and identified a transcriptional mechanism in which FGF-2 inhibits the expression of collagen I [14]. In our work, we observed that IGF1 and FGF2 have an inhibitory effect on the expression of collagen type I, compared to the medium without supplement. IGF1 is a factor that plays an important role in cartilage physiology [15] and has been shown to stimulate the synthesis of proteoglycans, as well as the proliferation and differentiation of mesenchymal cells; in addition, it is involved in the regulation of growth promoted by GH on cartilage and bone [16]. Tyler JA and Jenniskens et al. included bovine chondrocytes in alginate beads and added IGF-I. They observed an increase in the deposition of collagen II and proteoglycan, while type I collagen was deposited only on the surface. The formation of these two proteins was analyzed using IHC [11,17]. In this study, we report the presence of collagen II in large areas of the implant under all stimulus conditions. The most obvious effect was the combination of growth factors. Veilleux et al. included canine chondrocytes in a matrix of collagen/GAGs and added FGF-2 and/or IGF. They reported a greater deposition of collagen type II and GAGs through FGF2 stimulation [18]. However, tests were conducted under static culture conditions. Takahasi et al. tested 12 growth factors to evaluate the proliferation of human chondrocytes. The effect was more favorable between IGF1 and FGF2 [19]. FGF2 not only increases cell proliferation in monolayer during expansion but also promotes the redifferentiation potential and response to growth factors for a three-dimensional culture [20]. It has been demonstrated that FGF2 promotes the proliferation and synthesis of GAGs in chondrocytes cultured in monolayer [21]. Glycosaminoglycans are negatively charged in the sulfate carboxylated and can attract Ca^++^ and Na^++^ ions, a property which gives them a great ability to attract water and, therefore, absorb loads; this means that the forces come partially dissipated to the collagen network that is ultimately responsible for the three-dimensional structure of cartilage [22]. In our tests, we could detect a progressive increase in the GAGs, suggesting the possibility of a functional tissue due to the properties of these macromolecules to retain water, combined with an increase in the proportion of collagen II and a decrease in collagen I. The effect of motion in the cultures that were supplemented or not transfected increased the presence of all proteins analyzed in this study. The spinner flask was a system capable of maintaining the movement of explants with a small volume of culture medium; plus, various experiments were performed simultaneously. This method was generally allowed since the first week of culture and for all conditions, determining the presence of collagen fibers around the chondrocytes and glycosaminoglycans extending through the fibrin matrix.

## 5. Conclusions

Expression systems provide advantages by allowing continuous exposure to growth factors and cell receptors. However, recombinant proteins produce a similar effect of expression vectors when added to the culture medium constantly. While the continuous expression of growth factors suggests a greater production of GAGs, we found no difference in genetically manipulated cells or factors added to the environment. As such, we think the dynamic stimulus exerts an effect that allows the availability of nutrients situated in the culture medium for the chondrocytes trapped in the fibrin matrix, which facilitates their viability and protein synthesis. However, it requires growth factors that down-regulate or up-regulate the expression of extracellular matrix compounds.

## Figures and Tables

**Figure 1 polymers-16-01968-f001:**
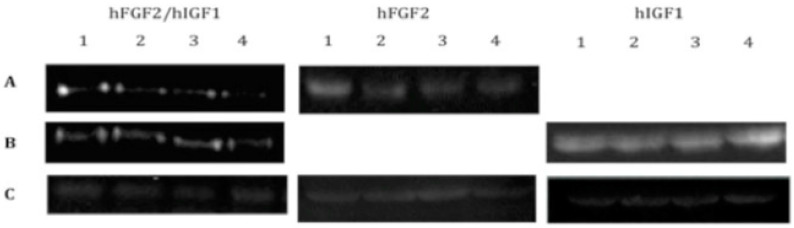
Western blots for the detection of growth factors lyophilized or secreted into the media by chondrocytes transfected with vectors expressing hFGF2, hIGF1, or hFGF2/hIGF1 and included in a fibrin-based, three-dimensional matrix. (**A**) Anti-hFGF2. (**B**) Anti-hIGF1. (**C**) Anti-actin.

**Figure 2 polymers-16-01968-f002:**
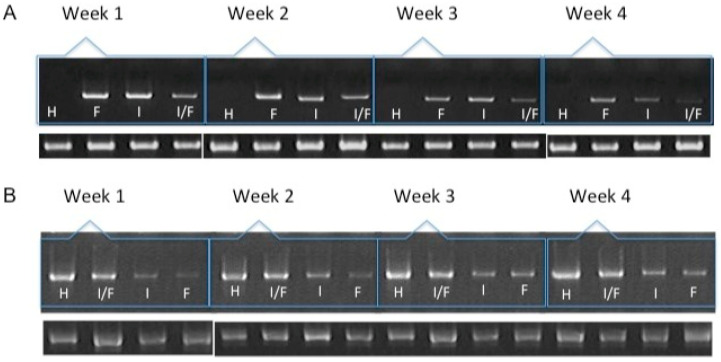
mRNA expression of collagen type I (**A**) and collagen type II (**B**) over the course of four weeks by chondrocytes transfected and included in a fibrin-based, three-dimensional matrix. H: hyaline cartilage. F: Chondrocytes transfected with pCIneo/hFGF2. I: Chondrocytes transfected with pCMVSport6/hIGF1. I/F: Chondrocytes co-transfected with plasmids expressing hFGF2 and hIGF1. The lower bands represent the expression of the constitutively expressed gene GAPDH.

**Figure 3 polymers-16-01968-f003:**
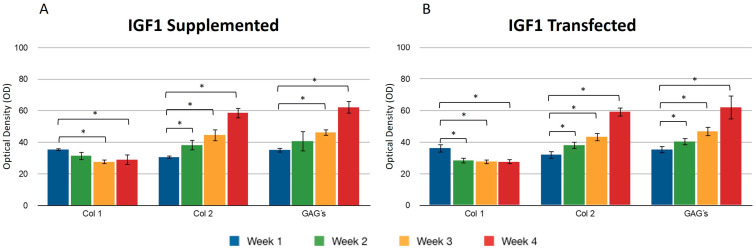
Supplemented and transfected IGF1. Densitometric analysis of positive areas throughout the three-dimensional fibrin matrix by the effect of IGF1. The Y-axis represents the OD measurements from densitometric analysis. Control samples were chondrocytes in a fibrin matrix without any growth factor supplementation (**A**) Expression of collagen type I, II, and GAGs over the course of four weeks under the stimulus of recombinant IGF1 added to the culture medium. (**B**) Expression of collagen type I, II, and GAGs over the course of four weeks under the stimulus of transfected IGF1. Col 1: collagen type I; Col 2: collagen type II; GAGs: glycosaminoglycans. Data are presented as means ± SD (n = 3). Statistical significance was determined using one-way ANOVA followed by Tukey’s post hoc test. * *p* < 0.05.

**Figure 4 polymers-16-01968-f004:**
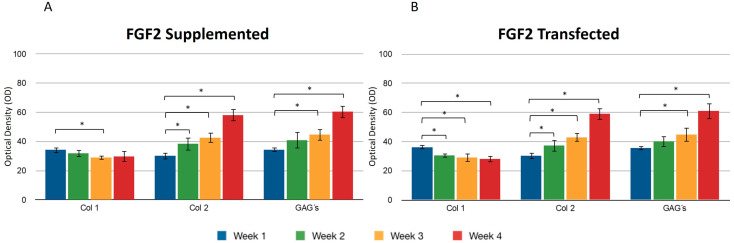
Supplemented and transfected FGF2. Densitometric analysis of positive areas throughout the three-dimensional fibrin matrix by the effect of FGF2. Control samples consisted of chondrocytes in a fibrin matrix without growth factor supplementation. (**A**) Expression of collagen type I, II, and GAGs over the course of four weeks under the stimulus of recombinant FGF2 added to the culture medium. (**B**) Expression of collagen type I, II, and GAGs over the course of four weeks under the stimulus of transfected FGF2. Col 1: collagen type I; Col 2: collagen type II; GAGs: glycosaminoglycans. Data are presented as means ± SD (n = 3). Statistical significance was determined using one-way ANOVA followed by Tukey’s post hoc test. * *p* < 0.05.

**Figure 5 polymers-16-01968-f005:**
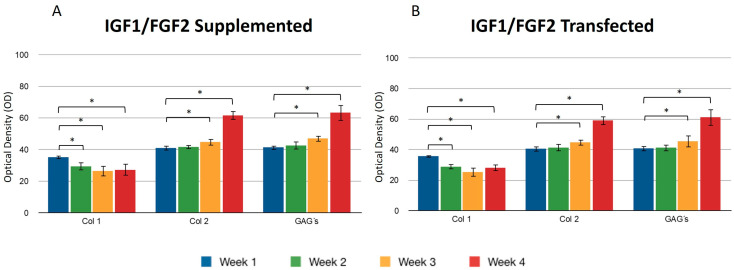
Supplemented and transfected IGF1/FGF2. Densitometric analysis of positive areas throughout the three-dimensional fibrin matrix by the effect of IGF1/FGF2. (**A**) Expression of collagen type I, II, and GAGs over the course of four weeks under the stimulus of recombinant IGF1/FGF2 added to the culture medium. (**B**) Expression of collagen type I, II, and GAGs over the course of four weeks under the stimulus of transfected IGF1/FGF2. Col 1: collagen type I; Col 2: collagen type II; GAGs: glycosaminoglycans. Data are presented as means ± SD (n = 3). Statistical significance was determined using one-way ANOVA followed by Tukey’s post hoc test. * *p* < 0.05.

**Figure 6 polymers-16-01968-f006:**
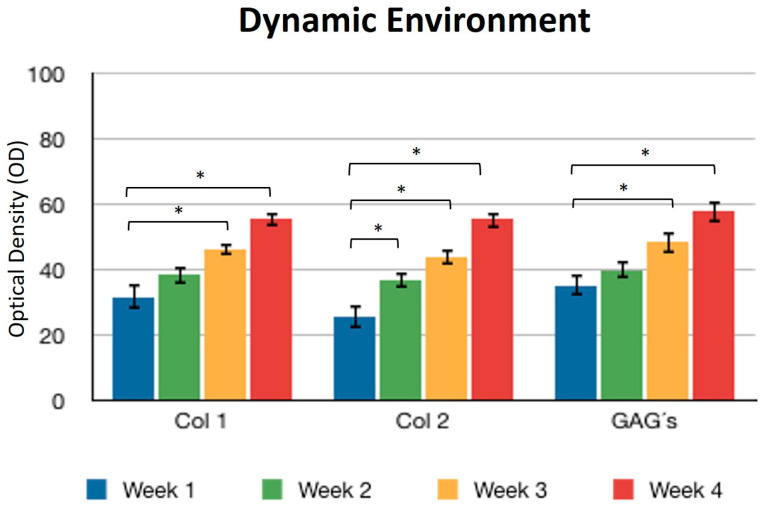
Dynamic environment. Expression of collagen type I, II, and GAGs over the course of four weeks by stimulation of a culture medium under constant agitation. Col 1: collagen type I; Col 2: collagen type II; GAGs: glycosaminoglycans. Data are presented as means ± SD (n = 3). Statistical significance was determined using one-way ANOVA followed by Tukey’s post hoc test. * *p* < 0.05.

**Figure 7 polymers-16-01968-f007:**
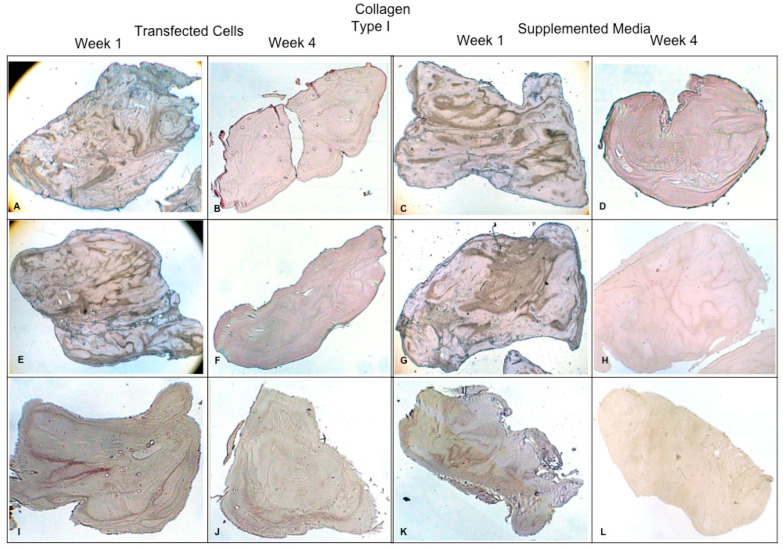
Fibrin explants included with chondrocytes that were added or transfected with IGF1 and/or FGF2. A specific antibody was used for detection of type I collagen fibers. The images show the explants at 1 week and 4 weeks. (**A**,**B**) hFGF2-transfected cells; (**C**,**D**) hFGF2-supplemented media; (**E**,**F**) hIGF1-transfected cells; (**G**,**H**) hIGF1-supplemented media; (**I**,**J**) hFGF2/hIGF1-transfected cells; (**K**,**L**) hFGF2/hIGF1-supplemented media.

**Figure 8 polymers-16-01968-f008:**
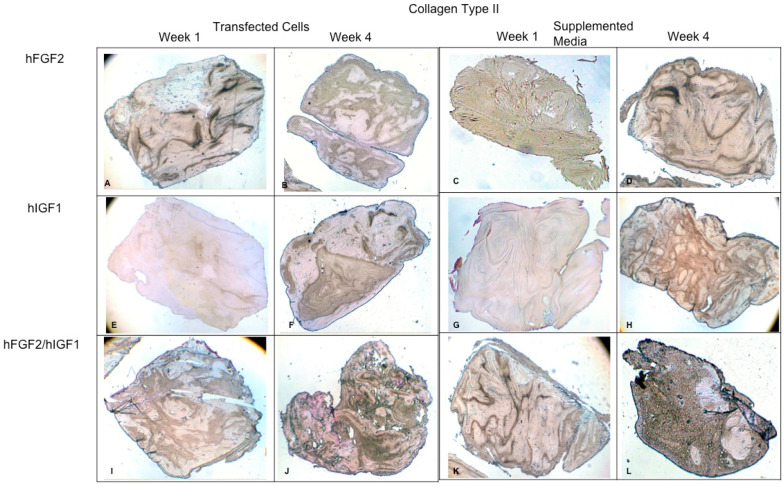
Fibrin explants included with chondrocytes that were added or transfected with IGF1 and/or FGF2. A specific antibody was used for detection of type II collagen fibers. The images show the explants at 1 week and 4 weeks. (**A**,**B**) hFGF2-transfected cells; (**C**,**D**) hFGF2-supplemented media; (**E**,**F**) hIGF1-transfected cells; (**G**,**H**) hIGF1-supplemented media; (**I**,**J**) hFGF2/hIGF1-transfected cells; (**K**,**L**) hFGF2/hIGF1-supplemented media.

**Figure 9 polymers-16-01968-f009:**
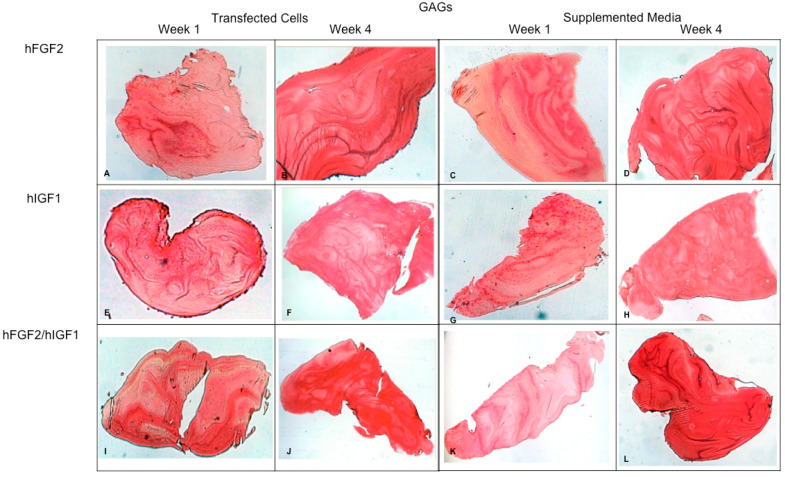
Fibrin explants included with chondrocytes that were added or transfected with IGF1 and/or FGF2. Histological staining with Safranin O was performed to detect the presence of GAGs. The images show the explants at 1 week and 4 weeks. (**A**,**B**) hFGF2-transfected cells; (**C**,**D**) hFGF2-supplemented media; (**E**,**F**) hIGF1-transfected cells; (**G**,**H**) hIGF1-supplemented media; (**I**,**J**) hFGF2/hIGF1-transfected cells; (**K**,**L**) hFGF2/hIGF1-supplemented media.

## Data Availability

The original contributions presented in the study are included in the article, further inquiries can be directed to the corresponding author/s.

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
