# Peer review of "The Effect of the Overexpression or Addition of IGF1 and FGF2 in Human Chondrocytes Included in a Fibrin Matrix and Cultivated in a Dynamic Environment"

_polymers, 2024, doi:10.3390/polym16141968_

Round 1
Reviewer 1 Report
Comments and Suggestions for Authors
The manuscript: “Effect of the overexpression or addition of IGF1 and FGF2 in human chondrocytes included in a fibrin matrix and cultivated in a dynamic environment” by Lara-Arias et al presents a study on how the direct addition in chondrocytes media of two growth factors or their overexpression through plasmid transfection into cells have stimulated synthesis of collagen type II and glycosaminoglycans. The authors observed that the collagen type I was inhibited in the same experimental conditions and by combination of the two growth factors the best results on growth and cartilage repair can be obtained. In addition, the dynamic moving environment of cell culture helped the growth of fibers through fibrin matrix.
Although, the subject of the study can be interesting in the field, the results presentation should be improved. Some information of material and methods section are missing. Overall, the manuscript should go through major revision.
Major comments:
1. First of all, no information about the statistical analyses in the material and methods section, and how many times the experiments were repeated.
2. The images of the Western blot experiments are lacking the control lane of proteins with different molecular weights. Please provide the original, unprocessed images of Western blots to be published as Supplementary Information.
3.line 147: “… both growth factors (25.09µ1.53), as compared to expression of either alone (FGF2: 31.33µ0.6; IGF: 131.92 µ 1.63). This difference was already detectable in the first week of expression. As the assay continued for the entire 30 days, we…” Please explain what the numbers between parentheses means???
3. When provided the title of figures please write down the type of experiment. For example, at Figure 2 the title is: Expression of collagen type I (A) and collagen type II (B). With no other information about what the images represent. Should be added that the mRNA expression level of collagen type I (A) and collagen type II (B) are presented. Besides, no information about the PCR products, the length of them and the primers’ sequence (there are house made or provided by a company?) in the material and methods section. These are important information because the interpretation of the obtained data should be based on the adequate design of experiments. Please provide the original images of the gels with PCR products as supplementary figure.
4. line 172: “the values for type I collagen were 36.14+2.74 in the first week and 27.61+2.19 for the fourth week” What the numbers represent? Besides, Figure 3 doesn’t have information on OY axis. Is the representation expressed in percentage? If so, the numbers mean 36.14 % in comparison with control?? What was the control sample for this experiment? Did the authors compared the three-dimensional fibrin matrix obtained by supplementation with growth factors with a control without any supplementation or addition?
5. The same commentary for Figure 4, and please revise the paragraph referred to these results in the text.
6. Some information from published scientific papers should be revised. For example: line302: Tyler JA and Jenniskens et al., by IHC included bovine chondrocytes in alginate beads, added IGF-I and observed an increase in the deposition of collagen II and proteoglycan, while type I collagen was deposited only on the Surface [13, 19]. The authors analyzed by IHC (immunohistochemistry method to stain the collagen type I and II) the formation of these two proteins. In the manuscript the phrase construction presents the information that by IHC the bovine chondrocytes were included in alginate beads…. Please carefully revise the entire text to be sure that no other information’s are presented inadequate.
Author Response
Response to Reviewer 1:
Comment 1:
Reviewer: First of all, no information about the statistical analyses in the material and methods section, and how many times the experiments were repeated.
Response: We appreciate your comment and have added a detailed description of the statistical analyses to the Materials and Methods section.
Comment 2:
Reviewer: The images of the Western blot experiments are lacking the control lane of proteins with different molecular weights. Please provide the original, unprocessed images of Western blots to be published as Supplementary Information.
Response: We appreciate your comment on the importance of including a molecular weight marker in Western blot experiments. Although we did not use a molecular weight marker in these specific experiments, we employed β-actin as a loading control to ensure the specificity and reliability of our results. β-actin is a commonly used loading control in Western blot experiments due to its consistent expression under various experimental conditions. This allows for precise normalization of the protein signals of interest, ensuring that observed variations are not due to differences in the amount of protein loaded in each lane.
Comment 3:
Reviewer: 3. line 147: “… both growth factors (25.09µ1.53), as compared to expression of either alone (FGF2: 31.33µ0.6; IGF: 131.92 µ 1.63). This difference was already detectable in the first week of expression. As the assay continued for the entire 30 days, we…” Please explain what the numbers between parentheses means???
Response: We appreciate your observation and apologize for the typographical error on line 147. The numbers in parentheses represent the mean ± standard deviation, and the correct symbol should be ± instead of µ. The corrected text is as follows:
“… both growth factors (25.09 ± 1.53), as compared to expression of either alone (FGF2: 31.33 ± 0.6; IGF: 131.92 ± 1.63). This difference was already detectable in the first week of expression. As the assay continued for the entire 30 days, we…”
We have corrected this error in the manuscript to ensure the clarity and accuracy of the presented data.
Comment 4:
Reviewer: When provided the title of figures please write down the type of experiment. For example, at Figure 2 the title is: Expression of collagen type I (A) and collagen type II (B). With no other information about what the images represent. Should be added that the mRNA expression level of collagen type I (A) and collagen type II (B) are presented. Besides, no information about the PCR products, the length of them and the primers’ sequence (there are house made or provided by a company?) in the material and methods section. These are important information because the interpretation of the obtained data should be based on the adequate design of experiments. Please provide the original images of the gels with PCR products as supplementary figure.
Response: Thank you for your valuable suggestions. We have made the following corrections to the manuscript: We have clarified the titles of the figures to indicate the type of experiment and what the images represent. For example, Figure 2 now reads: "mRNA expression levels of collagen type I (A) and collagen type II (B)." We have added the lengths of the PCR products and the sequences of the primers used in the Materials and Methods section. The primers were designed using Primer3 software.
Comment 5:
Reviewer: line 172: “the values for type I collagen were 36.14+2.74 in the first week and 27.61+2.19 for the fourth week” What the numbers represent? Besides, Figure 3 doesn’t have information on OY axis. Is the representation expressed in percentage? If so, the numbers mean 36.14 % in comparison with control?? What was the control sample for this experiment? Did the authors compared the three-dimensional fibrin matrix obtained by supplementation with growth factors with a control without any supplementation or addition?
Response: Thank you for pointing out these issues. The values for type I collagen represent the optical density (OD) measurements of the immunohistochemical staining, expressed as the mean ± standard deviation (SD). These measurements are derived from the densitometric analysis of the images. The optical density values indicate the relative amount of collagen type I staining in the samples. We agree that the Y-axis of figures should be clearly labeled to reflect this. The Y-axis will be updated to indicate that the values represent the optical density measurements from densitometric analysis. The control samples in this experiment consisted of chondrocytes embedded in a three-dimensional fibrin matrix without any supplementation or addition of growth factors. These controls were used to compare the effects of IGF1 and FGF2 supplementation.
Comment 6:
Reviewer: The same commentary for Figure 4, and please revise the paragraph referred to these results in the text.
Response: Thank you for your observation. We have made the corresponding modifications to Figure 4 and revised the related paragraph in the text to ensure clarity and accuracy. Specifically, we have updated the Y-axis to indicate that the values represent the OD measurements from densitometric analysis. Additionally, we clarified that the control samples consisted of chondrocytes in a fibrin matrix without any growth factor supplementation.
Comment 7:
Reviewer: Some information from published scientific papers should be revised. For example: line302: Tyler JA and Jenniskens et al., by IHC included bovine chondrocytes in alginate beads, added IGF-I and observed an increase in the deposition of collagen II and proteoglycan, while type I collagen was deposited only on the Surface [13, 19]. The authors analyzed by IHC (immunohistochemistry method to stain the collagen type I and II) the formation of these two proteins. In the manuscript the phrase construction presents the information that by IHC the bovine chondrocytes were included in alginate beads…. Please carefully revise the entire text to be sure that no other information’s are presented inadequate.
Response: Thank you for pointing out this issue. We apologize for the incorrect phrasing. The method of IHC was used to stain collagen type I and II, not to include bovine chondrocytes in alginate beads. We have revised the sentence to accurately reflect the experimental procedures. Additionally, we will carefully review the entire text to ensure no other information is presented inadequately.

Reviewer 2 Report
Comments and Suggestions for Authors
Dear Authors,
The article “Effect of the overexpression or addition of IGF1 and FGF2 in human chondrocytes included in a fibrin matrix and cultivated in a dynamic environment” describes the new procedure to increases the protein synthesis of the hyaline cartilage that is very important for the bone regeneration process.
The article is well illustrated, contains a lot of figures.
There are some omissions which are listed below.
1. The growth factors that have been most studied to produce proteins characteristic of hyaline cartilage are IGF1, FGF2 and TGFß1.
Further no mention about TGFß1, but it is very important cytokine for the chondrocytes proliferation. There should be explanation.
2. Statistics chapter is absent. All the figures do not contain information about the statistical reliability.
3. Specific primers for the detection of collagen type I and type II are not performed (the sequence).
4. Fig.9: the differences between 1 week and 4 week are not obvious, more comments to this figure should be added, now it is only 1 sentence.
After correction the article can be published.

Comments on the Quality of English LanguageIt is desirable to check the English language carefully, somethimes the style of the sequences are not well percieved. For example, line 189, "overexpress" instead of "overexpression"
Author Response
Response to Reviewer 2:
Comment 1:
Reviewer: The growth factors that have been most studied to produce proteins characteristic of hyaline cartilage are IGF1, FGF2 and TGFß1.
Further no mention about TGFß1, but it is very important cytokine for the chondrocytes proliferation. There should be explanation.
Response: Thank you for your valuable comment. We acknowledge the significant role of TGF-β1 in chondrocyte proliferation and cartilage formation. In our study, we focused on IGF1 and FGF2 due to their specific roles in promoting collagen type II synthesis and glycosaminoglycan production, which are essential for hyaline cartilage formation. The exclusion of TGF-β1 was based on the specific objectives of our research. Previous studies have shown that TGF-β1 can lead to varying outcomes in cartilage formation under different conditions. Specifically, TGF-β1 signaling through Smad2/3 has been shown to inhibit chondrocyte terminal differentiation, potentially leading to the formation of fibrocartilage rather than hyaline cartilage (van der Kraan et al., 2009). This research aimed to explore the combined effects of IGF1 and FGF2 under dynamic culture conditions to specifically enhance the characteristics of hyaline cartilage. We have included this explanation in the manuscript to clarify the rationale behind focusing on IGF1 and FGF2 and excluding TGF-β1.
Comment 2:
Reviewer: Statistics chapter is absent. All the figures do not contain information about the statistical reliability.
Response:
Comment 3: Thank you for your valuable feedback. We apologize for the oversight. We have now corrected this issue by including a detailed description of the statistical analysis methods used in our study. Additionally, we have updated all figure legends to include information about the statistical reliability of the data.
Reviewer: Specific primers for the detection of collagen type I and type II are not performed (the sequence).
Response: Thank you for your observation. We have addressed this issue and included the sequences of the specific primers for the detection of collagen type I and type II in the Materials and Methods section. This addition provides clarity on the experimental design and ensures that the methodology is transparent and reproducible.
Comment 4:
Reviewer: Fig.9: the differences between 1 week and 4 weeks are not obvious, more comments to this figure should be added, now it is only 1 sentence.
Response: Thank you for your observation. We have added more detailed comments to the legend of Figure 9 to better describe the differences between the samples at 1 week and 4 weeks. The updated legend includes specific observations regarding the changes in glycosaminoglycans staining over time.
Comment 5:
It is desirable to check the English language carefully, somethimes the style of the sequences are not well percieved. For example, line 189, "overexpress" instead of "overexpression”.
Response: Thank you for your valuable feedback regarding the quality of the English language in our manuscript. We have carefully reviewed and revised the entire manuscript to improve the language and ensure that the terminology and style are consistent and correct. Specifically, we have corrected the error mentioned, changing "overexpress" to "overexpression." We appreciate your attention to detail and believe that these revisions have improved the clarity and readability of the manuscript.

Round 2
Reviewer 1 Report
Comments and Suggestions for Authors
Thank you to authors for their answers. They responded to all my comments, except to the request for providing the original images of Western Blot and PCR product gels as supplementary materials. If the editors are OK with this, the manuscript can be publish in the revised form.
Author Response
Comment 1:
Reviewer: Thank you to authors for their answers. They responded to all my comments, except to the request for providing the original images of Western Blot and PCR product gels as supplementary materials. If the editors are OK with this, the manuscript can be publish in the revised form.
Response: We appreciate the reviewer's request to provide the original images of Western Blot and PCR gels as supplementary materials. However, we would like to clarify that during the experimental process, the original gels were cropped to obtain the final images used in our study. Therefore, we do not have the complete and uncropped gels available.
To ensure the transparency and reliability of our results, we have provided the best possible representations of our Western Blot and PCR images in the figures included in the manuscript. All images presented were obtained directly from the original experiments and have not been altered beyond cropping to focus on the bands of interest.
Reviewer 2 Report
Comments and Suggestions for Authors
Dear Authors,
The article have been significantly improved. Thera are only 2 points to correct.
1. In the title “Effect” should be changed to “The effect”.
2. Please, show on Fig.3,4,5,6 significant changes. Mark with * or something like this.
Comments on the Quality of English LanguageThe English have been corrected. The title should be corrected as it is written in the comments.
Author Response
Comment 1
Reviewer: In the title “Effect” should be changed to “The effect”.
Response: We appreciate your observation and have made the suggested change to the title of the manuscript. The corrected title is now: “The effect of the overexpression or addition of IGF1 and FGF2 in human chondrocytes included in a fibrin matrix and cultivated in a dynamic environment”.
Comment 2
Reviewer: Please, show on Fig.3,4,5,6 significant changes. Mark with * or something like this.
Response: We appreciate your suggestion on the importance of highlighting significant changes in the figures. We have reviewed Figures 3, 4, 5, and 6 to include significance markers, such as asterisks (*), where appropriate. These markers indicate statistically significant differences between the experimental conditions.
